# NAA10 (N-Alpha-Acetyltransferase 10): A Multifunctional Regulator in Development, Disease, and Cancer

**DOI:** 10.3390/cells14120863

**Published:** 2025-06-07

**Authors:** Zeng Quan Yang, Ion John Campeanu, Ivan Lopez, Manaal Syed, Yuanyuan Jiang, Hilda Afisllari

**Affiliations:** 1Molecular Therapeutics Program, Barbara Ann Karmanos Cancer Institute, Detroit, MI 48201, USA; 2Department of Oncology, Wayne State University School of Medicine, Detroit, MI 48201, USA; icampean@med.wayne.edu (I.J.C.); lopeziv1@udmercy.edu (I.L.); mssyed@wayne.edu (M.S.); jiangy@karmanos.org (Y.J.); hi6189@wayne.edu (H.A.)

**Keywords:** NAA10, acetyltransferase, N-terminal acetylation, developmental disorder, cancer

## Abstract

NAA10 (N-alpha-acetyltransferase 10) is a pivotal enzyme in eukaryotic cells, serving as the catalytic subunit of the NatA complex, which is responsible for the N-terminal acetylation of approximately 40–50% of the human proteome. Beyond its canonical role in co-translational N-terminal acetylation, NAA10 also acetylates internal lysine residues of various proteins and exerts non-catalytic regulatory functions through diverse protein–protein interactions. Pathogenic variants in *NAA10* are linked to a spectrum of developmental disorders, most notably Ogden syndrome, which is characterized by neurodevelopmental delay, cardiac defects, and craniofacial anomalies. In cancer, NAA10 is frequently overexpressed or genomically amplified, where its dysregulation correlates with tumor aggressiveness and poor prognosis. Functional studies implicate NAA10 in regulating cell cycle progression, apoptosis, migration, and other hallmarks of cancer. In this review, we summarize the structure, molecular mechanisms, and physiological functions of NAA10, as well as its roles in human diseases and cancer. We present in silico pan-cancer analyses that highlight its clinical significance and potential downstream pathways. Furthermore, we discuss the therapeutic potential and challenges of targeting NAA10 in cancer, and propose future research directions to better understand its multifaceted roles in health and disease.

## 1. Introduction

N-terminal acetylation, the enzymatic transfer of an acetyl group to the α-amino group of a protein’s N-terminal residue, is a highly prevalent post-translational modification in eukaryotes [1,2,3]. Affecting approximately 80–90% of human proteins, N-terminal acetylation influences protein folding, localization, complex formation, and degradation, as well as diverse cellular processes ranging from apoptosis to gene regulation [1,2,3,4,5,6,7,8,9]. The enzymatic machinery responsible for N-terminal acetylation comprises a subfamily of N-terminal acetyltransferases (NATs) belonging to the GCN5-related N-acetyltransferase (GNAT) family, which utilizes acetyl-CoA as the donor molecule for acetylation [1,2,3]. Eukaryotes possess eight NAT complexes (NatA–NatH), with humans expressing seven; NatG appears to be restricted to the plant kingdom [1,3,4]. These multimeric enzymes typically consist of a catalytic subunit partnered with one or two auxiliary components that modulate substrate specificity and cellular localization [1,2,3,4,5,6]. The substrate specificity of NATs is primarily determined by the N-terminal sequence of the substrate, particularly the first two amino acids of the polypeptide chain [1,10,11,12]. To date, no deacetylase-targeting acetylated N-termini has been identified, supporting the assumption that N-terminal acetylation is irreversible [4]. Importantly, evidence from model organisms and human genetic studies reveals that NAT-mediated acetylation critically regulates cellular homeostasis, embryonic development, and disease pathogenesis through both enzymatic and structural mechanisms [1,2,3,4,5,13,14].

Among the seven human NATs, NatA was the first to be identified and is responsible for acetylating approximately 40–50% of the human proteome [1,2,3,4,5,6,7,8,12,15,16]. The NatA complex consists of two core subunits: the catalytic subunit NAA10 (N-alpha-acetyltransferase 10, also known as ARD1, arrest-defective protein 1) and the auxiliary subunit NAA15, along with the regulatory subunit HYPK (Huntingtin-interacting protein K) [1,2,3,4,5,17]. The human genome also encodes paralogs of NAA10 and NAA15, known as NAA11 and NAA16, respectively, which can be incorporated into the NatA complex [18,19]. The NatA complex, with the broadest range of substrates, canonically targets small N-terminal amino acids, Ala, Ser, Thr, Val, Gly, and Cys, exposed following the cleavage of the initiator methionine by methionine aminopeptidases (MetAPs) [1,2,3,4,5,6]. Beyond its NAT activity, NAA10 also functions as a lysine acetyltransferase (KAT), capable of acetylating internal lysine residues on various proteins, although its KAT activity remains a subject of ongoing debate [20,21,22,23,24,25]. Over the past two decades, studies have established NAA10′s critical role in biological development, with mutations linked to developmental disorders [13,14,26,27,28]. In addition, accumulating evidence suggests that NAA10 can function either as an oncogene or a tumor suppressor, depending on the cancer type or cellular context [29,30,31,32,33,34,35,36]. In this review, we summarize the structure of NAA10, its molecular and biological functions in normal development, its involvement in human diseases, and its role in cancer progression. We present in silico analyses of The Cancer Genome Atlas (TCGA) and Clinical Proteomic Tumor Analysis Consortium (CPTAC) pan-cancer data to highlight the clinical significance and potential downstream targets of NAA10 in human cancers. Furthermore, we discuss the potential clinical implications and future prospects of NAA10 as a therapeutic target for cancer treatment.

## 2. Structural Basis and Molecular Functions of NAA10

The human *NAA10* gene, initially named *ARD1*, was first identified and cloned in 1994 by Tribioli et al. through a cDNA selection strategy targeting the Xq28 region of the X chromosome [16]. The human *NAA10* gene is composed of eight exons and is highly conserved across species, from yeast to mammals [3,16,37]. The human *NAA11* gene, also known as *ARD2*, is a paralog of *NAA10* and is located on chromosome 4q21 [18]. Previous studies, along with analyses of multiple transcriptomic and proteomic datasets, including the Genotype–Tissue Expression (GTEx) project and the Human Protein Atlas (HPA), have shown that NAA10 is broadly expressed across human tissues, whereas NAA11 exhibits tissue-specific expression, with enrichment in the placenta and testis [16,18,19,38,39]. Biochemical characterization by Arnesen et al. in 2005 identified NAA10 as the catalytic subunit of the NatA complex [40]. The NAA10 protein, consisting of 235 amino acids, contains an N-terminal NAT enzymatic domain (residues 45–130) and a C-terminal intrinsically disordered region (Figure 1A). NAA10 also contains a nuclear localization signal (NLS) spanning amino acids 78–83. The human NAA11 protein consists of 229 amino acids and shares 82% sequence identity with NAA10. Since *NAA11* is a tissue-specific gene predominantly expressed in the testis and is proposed to compensate for reduced levels of its X-linked paralog *NAA10* in this organ, this review will primarily focus on NAA10 [18,19].

Structural studies, mainly using X-ray crystallography and cryo-electron microscopy (cryo-EM), have elucidated the architecture of the NAA10 protein and its integration into the NatA complex [3,17,41,42,43,44]. The crystal structures of the human NatA complex, comprising NAA10 and the auxiliary subunit NAA15, with and without HYPK, were reported at resolutions of 3.15 Å and 2.80 Å, respectively (PDB: 6C95 and 6C9M) (Figure 1B) [41]. This study revealed that the structure of human NatA displays a high degree of structural conservation to the NatA structure of *Schizosaccharomyces pombe* [17,41]. The enzymatic domain of NAA10 adopts a GNAT fold, characterized by a central β-sheet (β4-β5) flanked by α-helices, forming a compact acetyl-CoA binding pocket [17,41,45]. NAA10 contains the acetyl-CoA binding motif (RxxGxG/A; RRLGLA in NAA10), a hallmark of GNAT enzymes, which coordinates acetyl-CoA through hydrogen bonding and hydrophobic interactions [46].

The NAA15 subunit is composed of more than 40 α-helices, which arrange themselves into a ring-like tertiary structure that wraps completely around the NAA10 subunit (Figure 1B) [3,17,41,42,45]. The most intimate interactions between NAA10 and NAA15 are made between several NAA15 helices and the N-terminal α1-loop-α2 segment of NAA10 [17,41]. By surrounding NAA10 in a ring-like manner, NAA15 remodels the NAA10 catalytic site for efficient catalysis [17,41,45]. Another component of the NatA complex, HYPK, was first identified as a protein that interacts with the Huntingtin protein [47,48]. HYPK is specific to higher eukaryotes, plants, and several fungi, but is absent in yeast [3,45]. HYPK interacts with numerous proteins, including NAA10, regulating NAA10 enzymatic activity [17,41,45,48]. HYPK binds to NatA via a bipartite mechanism: its ubiquitin-associated (UBA) domain interacts with a metazoan-specific region of NAA15, while its N-terminal loop-helix region distorts the NAA10 active site, regulating acetyltransferase activity [17,41,45].

Recent cryo-EM studies have provided insights into NAA10′s structure within multi-protein complexes on the human ribosome, highlighting its dynamic positioning for co-translational processing. Klein et al. (2024) reported two cryo-EM structures (PDB: 9FPZ, 9FQ0) of multi-enzyme complexes on vacant human 80S ribosomes, capturing NatA in complex with MetAPs [42]. This study demonstrated that human NatA occupies a non-intrusive “distal” binding site on the ribosome which does not interfere with MetAPs binding nor with most other ribosome-associated factors [42]. Lentzsch et al. further investigated NAA10′s role in a ribosomal multienzyme complex orchestrated by the nascent polypeptide-associated complex (NAC) [43,49]. NAC assembles a multienzyme complex with MetAP1 and NatA early during translation and prepositions the active sites of both enzymes for timely sequential processing of the nascent protein [43]. Notably, HYPK’s inhibitory role is modulated by NAC, which releases HYPK’s suppression of NAA10′s activity, allowing efficient acetylation of nascent chains [43].

NAA10 predominantly localizes to the cytoplasm, where it associates with ribosomes to mediate co-translational N-terminal acetylation of specific proteins [1,4,5,37]. It can also translocate to the nucleus, where it may regulate transcription through its KAT activity (Figure 1C) [50]. Previous studies have shown that NAA10 acetylates the ε-amino group of lysine residues on substrate proteins, including several transcription factors such as the androgen receptor (AR) and nuclear factor erythroid 2-related factor 2 (NRF2) [33,51,52,53]. Additionally, NAA10 exerts non-catalytic functions by interacting with proteins such as ADAM9 (ADAM metallopeptidase domain 9) and DNMT1 (DNA methyltransferase 1), to modulate their functions independently of acetylation [4,54,55,56,57,58].

In summary, NAA10 acts as a multifunctional regulator that may influence the folding, localization, complex formation, and degradation of various proteins, thereby modulating diverse cellular processes.

## 3. NAA10 in Developmental Disorders

### 3.1. Pathogenic Mutations and Developmental Roles of NAA10

Mutations in the *NAA10* gene have been linked to a spectrum of developmental disorders collectively referred to as *NAA10*-related syndromes, with Ogden syndrome (OMIM: 300855) being the most extensively studied [13,26,27,28,59,60,61]. In 2011, Rope et al. first reported a missense mutation in the *NAA10* gene (p.Ser37Pro) in eight boys from two unrelated families, presenting with global developmental delay, hypotonia, aged appearance, craniofacial anomalies, and cardiac arrhythmias [60]. This disorder was originally named Ogden syndrome, as suggested by the first family identified with the condition, in honor of their hometown, Ogden, Utah [26,60,61]. Biochemical studies demonstrated that the p.Ser37Pro variant had impaired NatA complex formation and reduced NAA10 catalytic activity [26,62]. Over the past decade, additional pathogenic variants in *NAA10*, including Arg4Ser, Tyr43Ser, Arg83Cys, Val107Phe, Arg116Trp, Phe128Leu/Ile, Met147Thr, have been identified in both males and females, with males predominantly affected due to hemizygous mutations in this X-linked gene [1,13,14,26,28,59,63,64,65]. The clinical features associated with pathogenic *NAA10* mutations are variable, but commonly reported features include developmental delay, intellectual disability, cardiac anomalies, brain abnormalities, facial dysmorphism, and visual impairment [1,13,14,26,28,59,63,64]. Moreover, a splice-site mutation in intron 7 of *NAA10* (c.471 + 2T > A) has been linked to Lenz microphthalmia syndrome, causing a range of developmental abnormalities and dysregulation of genes involved in embryonic and ocular development, particularly through the retinoic acid and Wnt signaling pathways [66].

The developmental roles of NAA10 have been investigated in various model systems, from yeast to mouse [15,21,37,57,67,68,69,70,71,72,73]. While NatA function is not essential for viability in *Saccharomyces cerevisiae*, deletion of *NAA10* (*naa10*Δ) or *NAA15* (*naa15*Δ) results in multiple physiological defects, including impaired mating and sporulation, delayed entry into the stationary phase, and increased sensitivity to temperature, salt, and various drugs [15,37]. In contrast, the NAA10 homolog is essential in several multicellular organisms. Its loss leads to lethality in *Drosophila melanogaster*, *Trypanosoma brucei*, and *Caenorhabditis elegans*, highlighting its critical role in development and cellular homeostasis [37,57,67,68,71].

Previous mouse studies have demonstrated that Naa10 is essential for neuronal development, bone formation, and spermatogenesis [14,51,74,75]. During brain development, mouse Naa10 is highly expressed in regions of active cell proliferation and migration, with expression decreasing as neurons differentiate [74]. Functional studies further reveal that Naa10 plays a critical role in osteoblast differentiation and early bone development [51]. Recent studies uncovered that Naa10 is not essential for mouse viability, as its paralog Naa12 largely compensates for the loss of Naa10, rescuing knockout mice from embryonic lethality [69,70]. However, the phenotypic variability in these *Naa10* knockout mice remains extensive, including piebaldism, skeletal abnormalities, reduced body size, hydrocephaly, hydronephrosis, and neonatal lethality [69,70]. Furthermore, analysis of data from the International Mouse Phenotyping Consortium (IMPC) revealed that *Naa10* knockout significantly impacts more than ten physiological systems, including the nervous, cardiovascular, hematopoietic, and visual systems [76]. Interestingly, a study using *Naa10*-knockout mice reported that Naa10 plays a role in fat metabolism and energy homeostasis by N-terminally acetylating Pgc1α (peroxisome proliferator-activated receptor gamma coactivator 1-alpha), a key regulator of mitochondrial biogenesis and liver gluconeogenesis [77,78,79]. The study also found that *NAA10* mRNA levels in adipose tissue positively correlate with obesity in both mice and humans [79].

In summary, NAA10 plays essential roles in the development of multiple organs, and mutations that reduce its enzymatic activity are associated with a wide spectrum of developmental defects.

### 3.2. Mechanistic Insights into NAA10-Related Developmental Syndromes

Since mutations in the *NAA10* gene are directly linked to human developmental disorders, NAA10 likely serves as a critical nexus in regulating fundamental biological processes. Molecular mechanisms of NAA10 in human diseases have been elucidated through detailed characterization of *NAA10*-related syndromes, particularly Ogden syndrome [1,14,26,61,62]. The *NAA10* p.Ser37Pro variant, first identified in Ogden syndrome, introduces a proline residue that disrupts the coiled-coil structure and reduces binding affinity with NAA15. This destabilizes the NatA complex and lowers its acetylation efficiency [62]. Studies have demonstrated that NAA15 in the NatA complex is essential for NAA10′s NAT activity but not for its KAT activity [1,17,63]. Unlike NAA10, NAA15 does not contain the NLS sequence; it is primarily cytoplasmic and functions to induce conformational changes and active site rearrangement of NAA10, as well as to anchor NAA10 to the ribosome for co-translational acetylation [17,43,63]. Recent genetic studies have identified mutations in the *NAA15* gene in dozens of individuals, leading to *NAA15*-related syndrome, which shares clinical similarities with *NAA10*-related syndrome despite exhibiting variable degrees of developmental disabilities [28,63,80,81,82]. Certain *NAA15* missense variants associated with *NAA15*-related syndrome, such as the p.Lys450Glu mutation, impair intermolecular interactions with NAA10 or HYPK, thereby affecting NatA catalytic activity or localization to the ribosome [28,63]. Thus, it is plausible to hypothesize that the majority of NAA10′s physiological functions are mediated through its role as the NatA catalytic subunit, which acetylates approximately half of the human proteome.

NAA10, along with other NATs, adds a hydrophobic and bulky acetyl group to the α-amino group of a protein’s N-terminal residue, altering the N-terminal characteristics of the protein (Figure 1C) [3]. N-terminal acetylation plays crucial roles in regulating protein folding and aggregation, modulating protein–protein interactions, controlling subcellular localization, and influencing protein stability and turnover [1,2,3,4,5,6,7,8]. Nevertheless, the direct protein targets of NAA10 and the mechanisms by which their acetylation contributes to the development and pathological features of *NAA10*-related syndromes remain poorly understood. Proteome-wide acetylation analysis from cellular models generated from males with Ogden syndrome revealed only a mild reduction in the in vivo acetylation of a small number of proteins [62]. These proteins include ELOA (elongin-A), GCN1 (GCN1 activator of EIF2AK4, also known as the stalled ribosome sensor GCN1), PPIA (peptidylprolyl isomerase A), RPL13A (ribosomal protein L13a), RPP30 (ribonuclease P/MRP subunit p30), and SAE1 (SUMO1 activating enzyme subunit 1), which were identified in both cellular models of Ogden syndrome and the NAA10-knockdown HeLa cancer model [37,62]. A newly published study using patient-derived induced pluripotent stem cell (iPSC) models has provided mechanistic insights into cardiomyopathy caused by NAA10 dysfunction [65]. In this study, Yoshinaga et al. (2025) used iPSC lines harboring the NAA10 p. (Arg4Ser) variant to profile the N-terminal acetylome, identifying 985 N-terminally acetylated proteins [65]. Gene Ontology Biological Process (GOBP) analysis revealed significant enrichment for proteins involved in metabolic pathways and RNA regulation/processing. This study also identified several key cardiac-associated proteins, such as MYH6 (myosin heavy chain 6), MYH7, TNNI3 (troponin I3, cardiac type), and DES (desmin), as N-terminally acetylated targets of NAA10 in their iPSC models [65]. Given the broad substrate pool of the NatA complex, the varying degrees of developmental disabilities observed in *NAA10*-related syndromes, and the functional consequences of *NAA10* mutations, are likely influenced by the specific mutation site and status, as well as the identity of the affected target proteins, rather than being solely attributable to the loss of NAA10′s acetylation activity. Furthermore, a study using a *Naa10*-knockout mouse model revealed an unexpected role for NAA10 in maintaining global DNA methylation by facilitating DNMT1 binding to DNA substrates [57]. This study demonstrated that the Ogden syndrome-causing *NAA10* mutation disrupts the binding of NAA10 and DNMT1 to imprinting control regions of the genome, suggesting that the non-catalytic functions of NAA10 also contribute to the development and pathological phenotypes of *NAA10*-related syndromes.

## 4. NAA10 in Cancer

### 4.1. Alterations and Clinical Significance of NAA10 in Human Cancer

Beyond its role in development, NAA10 has emerged as a significant player in cancer biology over the past two decades [29,30,31,32,33,34,35,36,53,56,58,83,84,85,86,87,88,89,90]. NAA10-mediated protein acetylation plays a crucial role in regulating key cellular processes relevant to cancer development, including cell cycle, migration, apoptosis, autophagy, differentiation, and proliferation. Mounting evidence indicates that NAA10 is overexpressed in various cancers, including breast, colon, esophageal, liver, lung, and prostate cancers [29,30,31,32,33,34,35,53,56,58,83,84,91,92]. This overexpression correlates with aggressive cancer phenotypes and poor prognosis, positioning NAA10 as a potential marker and therapeutic target.

Our group previously conducted a metagenomic analysis of 37 human acetyltransferases across more than 10,000 cancer samples from 33 TCGA tumor types, with a particular focus on breast cancer [31]. We found that *NAA10* is one of the most frequently amplified acetyltransferase genes in the TCGA dataset. In breast, ovarian, and uterine cancers, *NAA10* gene amplification rates exceeded 5% in specific subsets of these malignancies [31]. Additionally, we found that higher *NAA10* mRNA expression was significantly associated with tumor aggressiveness and poor prognosis in breast cancer patients. Loss-of-function analyses in breast cancer models further indicated that NAA10 plays a critical role in promoting cancer cell growth and survival [31].

Comparing mRNA and protein expression levels between tumor and normal tissues provides important insights into genes that may function as cancer drivers and potential therapeutic targets. To determine the expression patterns of NAA10 across a large cohort of cancer samples, we analyzed its expression changes in tumor versus normal tissues using data from the TCGA and CPTAC datasets [93,94,95,96,97]. We selected tumor types with at least ten normal tissue samples available for comparison. Among the 16 tumor types from the TCGA dataset that met this criterion, *NAA10* showed significantly elevated RNA expression levels in tumors (*p* < 0.05) in 15. Nine tumor types, including breast, colon, endometrial, liver, lung adenocarcinoma, and lung squamous cell carcinoma, showed log_2_-fold changes exceeding 0.5 (Figure 2A). Similarly, in the 12 tumor types analyzed from the CPTAC proteomics dataset, NAA10 exhibited significantly elevated protein expression levels (*p* < 0.05) in 11 cancer types compared to normal controls. Breast cancer and lung squamous cell carcinoma showed the greatest differential expression, with log_2_-fold changes greater than 0.5 (Figure 2B). Consistent with previous studies, analysis of NAA10 expression in the TCGA and CPTAC datasets also revealed that NAA10 mRNA or protein levels are substantially higher in tumors with advanced stage or grade in several cancer types, such as liver cancer in the TCGA dataset and lung adenocarcinoma in the CPTAC dataset [29,30,32,35,92].

Additionally, we analyzed immunohistochemistry (IHC) data using an anti-NAA10 antibody (HPA030711, Atlas Antibodies) across 190 tumor samples from 20 cancer types in the HPA. The IHC staining was evaluated based on (1) intensity (negative, weak, moderate, or strong), (2) fraction of stained cells (<25%, 25–75%, or >75%), and (3) subcellular localization (nuclear and/or cytoplasmic/membranous) [39,98]. In most tumor samples, NAA10 exhibited moderate cytoplasmic staining, with additional nuclear or nuclear membrane localization observed in some cases. Based on staining intensity and the fraction of stained cells, the IHC data were categorized into four groups: high, medium, low, and not detected. Among the 190 tumor samples, 17 (8.95%) showed a high NAA10 IHC score, 104 (54.74%) were medium, 40 (21.05%) were low, and 29 (15.26%) showed no detectable staining. Among the 20 tumor types, five—head and neck, pancreatic, prostate, urothelial cancers, and lymphoma—had more than 20% of samples showing high NAA10 staining. Representative images of two tumor samples with high NAA10 expression in lung and pancreatic cancers are shown in Figure 2C. In summary, data from CPTAC and HPA clearly demonstrate that NAA10 protein is highly expressed in a subset of primary human tumor samples.

Given that phosphorylation is a key regulatory mechanism influencing protein activity, including the acetyltransferase activity of NAA10, we also analyzed phosphoproteomics data from the CPTAC dataset to compare NAA10 phosphorylation between tumor and normal samples [93,95,99,100]. Remarkably, almost all phosphorylation sites identified in the CPTAC samples were localized to the C-terminal region of the NAA10 protein, particularly within its intrinsically disordered region. This strongly supports a regulatory role for the C-terminus of NAA10 under both physiological and pathological conditions. Among the more than 15 phosphorylation sites identified in the CPTAC dataset, five sites—S167, S171, S174, S190, and S194—exhibited increased phosphorylation levels, even after normalization to total NAA10 protein abundance, in several tumor types, including breast and lung cancers.

Our integrated analysis of large-scale cancer datasets, along with multiple published studies, demonstrates that NAA10 is highly expressed at both the RNA and protein levels across various cancer types, strongly supporting its role in promoting tumorigenesis [30,32,35,56,92]. However, earlier studies have suggested that NAA10 may function as a tumor suppressor in certain tumor types or cellular contexts [54,55,88]. To further clarify the role of NAA10 in human cancer, we analyzed genome-wide loss-of-function screening datasets across a broad panel of tumor cell lines representing multiple cancer types [101,102]. Accordingly, we queried shinyDepMap, a web-based tool that integrates CRISPR and shRNA screening data from the Cancer Dependency Map (DepMap) project [102]. The combined effect scores from both CRISPR and shRNA datasets across a panel of tumor cell lines provide two key metrics for each gene: the extent to which gene loss impairs cell growth in sensitive lines (“efficacy”) and the variability of its essentiality across different lines (“selectivity”) [102]. This means that the lower the efficacy score, the more essential the gene is for tumor cell growth and survival. Using the shinyDepMap tool, we analyzed seven human NAA genes, including all catalytic subunits of the seven human NAT complexes: *NAA1*0 (NatA), *NAA20* (NatB), *NAA30* (NatC), *NAA40* (NatD), *NAA50* (NatE), *NAA60* (NatF), and *NAA80* (NatH) [1,5]. Remarkably, *NAA10* had the lowest efficacy score (−1.837) and the highest selectivity score (0.393) among the seven human NAT genes, and it also ranks in the top 5% of the 15,847 human genes analyzed in shinyDepMap (Figure 3). We also recognized that *NAA10* exhibits efficacy and selectivity scores comparable to those of the classical oncogene *MYC* (−1.947 and 0.396, respectively), as highlighted in the shinyDepMap plot where the two genes are positioned in close proximity (Figure 3) [103]. These findings emphasize a strong dependency of cancer cells on NAA10, supporting its role as a potential oncogenic driver and a promising therapeutic target across diverse tumor types.

### 4.2. Multilayered Regulation of NAA10 in Cancer

The transcriptional and signaling regulation of NAA10 in cancer has been broadly investigated both in vitro and in vivo [1,29,30,33,34,35,61]. Since works by our group and others have shown that NAA10 is highly expressed at both the mRNA and protein levels in various cancers, we first investigated the cis-association among copy number variation, mRNA expression, and protein abundance of NAA10 using the CPTAC dataset [93,97]. We found that NAA10 protein abundance is significantly positively associated with copy number variation and mRNA expression in certain tumor types, notably lung cancer [93,94]. This data, aligned with previous studies, suggest that the elevated expression of NAA10 in certain tumors is, at least in part, due to copy number increases at the *NAA10* gene locus (Xq28 region) [31,83].

In addition, previous studies have shown that NAA10 expression and activity are regulated at multiple levels in human cancers, contributing to its oncogenic and/or tumor-suppressive roles depending on the cellular context. Transcriptionally, the classical oncogenic transcription factor MYC directly binds to the *NAA10* promoter and regulates its expression in esophageal cancer [83]. Interestingly, as illustrated in the shinyDepMap plot, *NAA10* and *MYC* display similar efficacy and selectivity scores in CRISPR and shRNA screens, further supporting a regulatory association between NAA10 and MYC in certain cancers (Figure 3). Post-transcriptionally, several microRNAs, including *miR-342-5p* and *miR-608*, suppress *NAA10* expression by targeting its 3′-untranslated region in colon cancer [104]. NAA10 activity is also post-translationally modulated through phosphorylation, auto-acetylation, and other modifications, primarily within its C-terminal region [29]. For instance, previous studies revealed that phosphorylation at S209 by IKKβ (inhibitor of κB kinase β) promotes proteasomal degradation of NAA10, whereas mTOR (mammalian target of rapamycin)-mediated phosphorylation at S228 enhances its acetylation activity [29,105,106,107]. A recent study on mouse Naa10 discovered that protein kinase C-delta interacts with and phosphorylates Naa10 in the N-terminal region, including at S80 and S108 [108]. Analysis of phosphoproteomics data from the CPTAC dataset revealed alterations in phosphorylation levels at several sites of NAA10 protein across multiple tumor types, supporting the regulation of NAA10 activity by kinases and/or phosphatases [95,99,100]. NAA10 can also auto-acetylate at lysine 136, promoting its functional activation [87]. Additionally, hormonal signals, such as androgens, can increase NAA10 expression in prostate cancer [86]. These multilayered regulatory mechanisms underscore the complex control of NAA10 expression and function in tumor progression.

### 4.3. Acetyltransferase-Dependent and -Independent Functions of NAA10 in Cancer

Similarly to NAA10-related syndromes, the biological roles of NAA10 in cancer likely involve both its acetylation-dependent mechanisms (including Nα- and Nε-acetyltransferase activities) and acetylation-independent mechanisms, depending on the cellular context [2,29,30]. Table 1 summarizes proteins that are either N-terminally or lysine-acetylated by NAA10, as along with proteins known to interact with NAA10, as identified in various human and mouse models. These NAA10-acetylated or -interacting proteins participate in diverse cellular pathways, including cell cycle regulation, migration, apoptosis, and autophagy, that collectively contribute to cancer cell proliferation, survival, and metastasis. It is worth mentioning that NAT10 (N-acetyltransferase 10), an RNA cytidine acetyltransferase, is distinct from the protein acetyltransferase NAA10 discussed in this review, although both enzymes play important but non-overlapping roles in cancer biology [109,110].

Functioning as a NAT, NAA10 has been shown to mediate N-terminal acetylation of a set of proteins identified in cancer cell line models [37,62,84,88,111]. An early proteomics analysis using siRNA-mediated knockdown of NAA10 in HeLa cells identified 16 proteins with altered N-terminal acetylation [37]. As mentioned earlier, combined analyses of Ogden syndrome cellular models and HeLa cancer models identified a subset of common N-terminally acetylated proteins, including GCN1, ELOA, PPIA, RPL13A, RPP30, and SAE1 [37,62]. Another early study in breast cancer revealed that NAA10 directly interacts with tuberous sclerosis complex 2 (TSC2) and mediates its N-terminal acetylation, resulting in TSC2 stabilization, suppression of mTOR signaling, and induction of autophagy [88]. Recently, a study showed that NAA10 promotes the invasion and metastasis of osteosarcoma cells by N-terminally acetylating matrix metallopeptidase 2 (MMP-2), thereby preventing its degradation [84]. More recently, proteomic analysis of the human epidermoid carcinoma cell line A431 following siRNA knockdown of both NAA10 and NAA15 identified 173 proteins with altered N-terminal acetylation, including GCN1, ALKBH7 (alpha-ketoglutarate-dependent dioxygenase ALKB homolog 7), and TIMM8B (translocase of inner mitochondrial membrane 8 homolog B) [111]. These NAA10 N-terminally acetylated proteins are involved in various biological pathways and processes, and their alterations may contribute to cancer growth and progression.

As a KAT, NAA10 acetylates internal lysine residues on various proteins identified in cancer models, including AR, AURKA (Aurora kinase A), CTNNB1 (β-catenin), HSP70 (heat shock protein 70), MLCK (myosin light-chain kinase, also known as MYLK), MSRA (methionine sulfoxide reductase A), NRF2, PGK1 (phosphoglycerate kinase 1), SAMHD1 (SAM domain and HD domain-containing protein 1), and NAA10 itself (Table 1). Lysine acetylation alters protein conformation and electrostatic properties by neutralizing positive charges, thereby affecting transcription factor binding, protein stability, and protein–protein interactions [21]. For example, NAA10 is overexpressed in prostate cancer, where it promotes cell growth in vitro and tumor formation in vivo by acetylating the AR at lysine 618, leading to activation of AR target genes [33,86,112]. In breast cancer, NAA10-mediated acetylation of AURKA at lysine 75 and 125 promotes the proliferation and migration of MCF-7 cells [113]. NAA10 regulates cell cycle progression, particularly at the G1/S transition, through the acetylation of substrate proteins such as β-catenin and CDC25A (M-phase inducer phosphatase 1) [89]. Studies have found that NAA10-mediated acetylation of HSP70 and the apoptosis regulator MCL1, two key regulators of apoptotic pathways, plays a critical role in promoting cell survival [114,115]. Furthermore, unlike NAA10-mediated N-terminal acetylation of TSC2, its lysine acetylation of PGK1 at lysine388, followed by Beclin1 phosphorylation, is required for autophagy induction and promotes glioblastoma progression [85,105,116].

Although NAA10 has been shown to function as a KAT in various studies, particularly in the context of cancer, this activity remains a subject of ongoing debate, as mentioned previously [20,21,22,23,24,25]. Structural analyses by Magin et al. (2016) revealed that the NAA10 active site is optimized for N-terminal acetylation, raising questions about its ability to efficiently accommodate internal lysine residues [22]. Additionally, in vitro biochemical assays using recombinant NAA10 failed to acetylate lysine residues within MSRA and RUNX2 [22]. Based on these findings, Magin et al. proposed that the observed lysine acetylation of MSRA and RUNX2 may result from non-enzymatic chemical reactions rather than direct enzymatic activity by NAA10 [22]. However, a study by Vo et al. (2020) demonstrated that recombinant human NAA10 can acetylate internal lysine residues in vitro, but only under specific, tightly controlled conditions [20]. Notably, the monomeric form of NAA10 exhibited KAT activity, whereas its oligomeric form, which accumulates over time during protein purification, lost this activity [20]. Furthermore, multiple studies using cancer cell models have shown that NAA10 acetylates specific lysine residues on various substrate proteins, as listed in Table 1. Nevertheless, while evidence supports a potential KAT function for NAA10, particularly in cancer models, further biochemical and structural studies are needed to definitively characterize its lysine acetylation activity, as well as its underlying structural and biochemical mechanisms.

Independent of its acetyltransferase activity, NAA10 also interacts with proteins such as ADAM9, DNMT1, PIX (p21-activated kinase interacting exchange factor), and STAT5A (signal transducer and activator of transcription 5A), regulating their stability and functional activity (Table 1). For instance, NAA10 forms a complex with ADAM9, stabilizing it and promoting prostate cancer invasiveness [58]. In vitro and in vivo studies have revealed the oncogenic role of NAA10 in lung cancer, demonstrating that NAA10 binds to DNMT1 and positively regulates its enzymatic activity by facilitating DNMT1′s binding to DNA and its recruitment to the promoters of tumor suppressor genes, such as E-cadherin [56]. Through its interaction with PIX or STAT5A, NAA10 has been shown to inhibit invasion and metastasis in breast cancer models [54,55]. Interestingly, NAA10′s effects on metastasis appear to be context-dependent; it may either inhibit or promote cancer cell motility depending on the specific acetylation targets and the signaling pathways involved [53,54,55,83]. This dual role emphasizes the need for further investigation to unravel the specific molecular contexts that dictate whether NAA10 serves as a pro- or anti-metastatic factor.

### 4.4. In Silico Identification of NAA10-Regulated Targets in Cancer

Proteomic profiling of large datasets from various cancer samples in the CPTAC project offers a valuable opportunity to investigate the correlation networks, signaling pathways, and potential downstream targets of cancer-related proteins [117]. A high correlation between proteins in CPTAC samples suggests structural or functional interconnection, such as being components of the same protein complex, being involved in shared signaling pathways, or representing direct downstream targets in normal and/or cancer cells [117]. Accordingly, we queried the correlation data of NAA10 with all other identified proteins in CPTAC samples and found that the three NatA complex components, NAA10, NAA15, and HYPK, are significantly positively correlated with each other in the CPTAC dataset [94]. For example, NAA10 and NAA15 proteins are significantly correlated in LUAD, with a Spearman correlation coefficient of 0.80 (*p* = 2.2 × 10^−16^). Similarly, another NAA10-interacting protein, DNMT1, shows a positive correlation in several CPTAC tumor types, with LUAD displaying a Spearman correlation coefficient of 0.48 (*p* = 2.2 × 10^−7^).

Since the major functions of NAA10 involve regulating protein folding, localization, complex formation, and degradation, changes in NAA10 protein abundance in cancer cells are likely to influence the levels of other proteins, some of which may be direct substrates of its acetyltransferase activity. Thus, we extracted the top 1082 proteins (729 positively correlated and 351 negatively correlated) that showed strong correlation with NAA10 protein levels, using a Meta *p*-value cutoff of 10 in the LinkedOmicsKB based on the CPTAC dataset [94]. Meta *p*-values were calculated using the “sumz” method from the R package to integrate correlation results across pan-cancer samples [93]. Subsequently, we performed functional and pathway analyses of the NAA10-positive and -negative proteins separately using the online Metascape tool, which combines functional enrichment, interactome analysis, gene annotation, and membership search to leverage over 40 independent knowledgebases within a single integrated portal [118]. Based on Metascape’s analysis, the top functional pathways of these positively correlated proteins include rRNA and mRNA metabolic processes, translation, the cell cycle, and others (Figure 4). In contrast, the top functional pathways of the negatively correlated proteins involve cytoskeleton and actin-related processes, focal adhesion, cell junctions, and others. Given the controversial role of NAA10 in cancer invasion and metastasis, particularly reports of its function as a metastasis suppressor in several studies, these negatively correlated proteins may represent downstream targets of NAA10 that contribute to its role in metastatic regulation. Nonetheless, further studies are necessary to test this hypothesis.

To narrow down potential NAA10-acetylated targets in cancer, we integrated the 1082 NAA10-correlated proteins with previously identified NAA10 substrates from numerous experimental models. First, we analyzed 12 previously reported proteins (excluding NAA10 itself) that were identified as internal lysine acetylation targets of NAA10 functioning as KAT (Table 1). Among these, only one protein, MLCK, showed a significant negative correlation with NAA10 protein abundance in the CPTAC dataset. An early study found that NAA10 binds to and acetylates MLCK at lysine 608, consequently inactivating MLCK. The inactivation of MLCK reduces the phosphorylation of myosin light chain (MLC), ultimately leading to the inhibition of tumor cell migration and invasion [119].

Second, we integrated our dataset with 985 N-terminally acetylated proteins identified from human Ogden iPSC models, which represent the most comprehensive set of NAA10 targets reported to date [65]. We found that 123 NAA10 positively correlated, and 46 negatively correlated, proteins overlapped with the 985 N-terminally acetylated proteins identified from human Ogden iPSC models. To further narrow down the most relevant and likely direct targets of NAA10 in cancer, we merged these 123 and 46 proteins with the 173 N-terminally acetylated proteins identified in A431 cancer cells following siRNA knockdown of both NAA10 and NAA15. By combining these three datasets, we identified a total of 21 proteins, 19 of which are NAA10-positively correlated, and 2 are NAA10-negatively correlated in CPTAC dataset (Table 2). While the two negatively correlated proteins are histone linker proteins (H1-0 and H1-4), the 19 NAA10 positively correlated proteins—including AIMP1 (aminoacyl-tRNA synthetase complex-interacting multifunctional protein 1), GCN1, PA2G4 (proliferation-associated 2G4), RPL11, and TIMM8B—play important roles in cell growth, translation, and RNA processing. Particularly, GCN1, RPL11, TIMM8B, and H1-4 were among the 34 proteins with altered N-terminal acetylation identified in wild-type and NAA15-mutant iPSCs [82]. More remarkably, GCN1 was also among the 16 proteins with altered N-terminal acetylation identified in an early study using HeLa cancer cells following NAA10 knockdown [37]. GCN1, RPL11, TIMM8B, and H1-4 were consistently found across multiple datasets, supporting their identification as direct N-terminally acetylated targets of NAA10 (Figure 5). Among them, GCN1, a protein involved in regulating protein synthesis and other cellular processes in response to amino acid starvation, stands out as a robust candidate and may serve as a valuable marker for investigating the functions, roles, and mechanisms of NAA10 in various cancer models.

## 5. Therapeutic Potential of Targeting NAA10

Given that NAA10 alterations directly contribute to human developmental disorders and cancers, it is critical to develop NAA10 inhibitors or other molecular tools to further investigate its functional roles and underlying mechanisms in these diseases, and to evaluate its potential as a therapeutic target. Despite the biological importance of NAA10, the development of specific small-molecule inhibitors remains limited. To date, only one study has reported a class of bisubstrate inhibitors targeting NAA10 [120]. Foyn et al. pioneered the design of CoA–peptide conjugates, which target both NAA10 and NAA50 by leveraging the substrate specificity of each enzyme [120]. These inhibitors utilize substrate-specific peptide motifs to achieve isoform selectivity, with one compound, CoA-Ac-EEE4, acting as a competitive inhibitor of NAA10′s acetyltransferase activity [120]. This study demonstrated the feasibility of pharmacologically targeting NAA enzymatic pockets, providing a foundation for the future development of more specific and potent inhibitors to dissect NAA10′s function in cancer and assess its therapeutic relevance.

As described previously, NAA10 belongs to the GNAT family within the larger KAT superfamily, which includes more than 30 members in humans, such as KAT2A, KAT2B, KAT6A, KAT6B, EP300, and CREBBP (CREB binding lysine acetyltransferase) [1,2,3,31,121,122]. These KATs are frequently altered in various human cancers, through mutations, translocations, gene amplification, or overexpression; therefore, inhibitors targeting them have been extensively investigated [31,121,123,124,125]. Over the past two decades, various approaches have been used to develop inhibitors targeting KAT2A (GCN5), but their potency remains suboptimal [125,126,127]. In 2017, Lasko et al. overcame the long-standing challenge of developing a drug-like KAT inhibitor by identifying A-485, a first-in-class, highly potent, selective, cell- and in vivo-active EP300/CREBBP catalytic inhibitor [128]. A-485 has been shown to inhibit the proliferation of various cancer cell lines, including those from prostate cancer, melanoma, multiple myeloma, and non-Hodgkin lymphoma [123,128,129,130]. More recently, catalytic inhibitors of KAT6A and KAT6B have been identified [125,131,132,133]. One such compound, PF-07248144, demonstrated a tolerable safety profile and durable antitumor activity in heavily pretreated estrogen receptor-positive (ER+), metastatic breast cancer in a Phase 1 trial (NCT04606446) [133]. These successfully identified KAT inhibitors, particularly those targeting KAT6, have established KATs as druggable targets in cancer. Although NAA10 shares the conserved GNAT fold with other GNAT enzymes, suggesting potential overlap in inhibitor binding, its unique role in the NatA complex, along with its dual NAT and debated KAT activities, reflects distinct substrate specificity and active site architecture compared to other GNATs or KATs such as KAT6 or EP300/CREBBP [41,45,121,122,123,125,127]. Nevertheless, further studies are needed to assess the cross-reactivity of currently available KAT catalytic inhibitors, and to explore the potential of structure-guided drug design for developing selective NAA10 inhibitors.

Targeting protein–protein interactions (PPIs) has become an increasingly attractive strategy for cancer drug discovery, as many oncogenic drivers and signaling complexes depend on PPIs that were previously considered “undruggable” by traditional small-molecule inhibitors. For example, inhibitors of the MDM2–p53 interaction, such as idasanutlin, are now in advanced stages of clinical development [134,135]. Previous studies have also demonstrated the therapeutic potential of inhibitors targeting KAT-involved PPIs. For example, small molecules that disrupt the EP300–MYB interaction inhibit acute myeloid leukemia (AML) cell proliferation both in vitro and in vivo [136,137]. Additionally, multiple EP300/CREBBP inhibitors targeting the bromodomain-mediated PPIs of EP300/CREBBP have been identified and clinically developed. Several of these compounds, including Inobrodib (CCS1477), FT-7051, and NEO2734, have entered clinical trials for the treatment of various cancers, such as leukemia, lymphoma, breast, lung, and prostate cancers [123,125,138,139,140]. Biochemical and structural studies have demonstrated that the interaction between NAA10 and NAA15 is critical for NatA complex activity [17,62]. Given detailed understanding of the interaction interface and key residues involved, it is plausible to hypothesize that inhibitors targeting the NAA10–NAA15 interaction could be identified and developed as a novel therapeutic strategy for cancer treatment. Furthermore, NAA10 also interacts with other proteins, such as DNMT1. FDA-approved DNMT1 inhibitors, including decitabine and azacitidine, are currently used to treat myelodysplastic syndromes; however, these drugs target the enzymatic activity of DNMT1 rather than its interaction with NAA10 [141,142]. In the future, it would be worthwhile to investigate whether these DNMT1 inhibitors have therapeutic potential in tumors with NAA10 overexpression.

Protein degraders, including PROTACs (Proteolysis Targeting Chimeras) and molecular glue degraders, represent a novel and promising therapeutic modality for cancer treatment [143,144]. By inducing proximity between a target protein and an E3 ubiquitin ligase, these degraders can eliminate both enzymatic and nonenzymatic proteins. Early clinical trials targeting the androgen receptor, estrogen receptor, and BTK (Bruton tyrosine kinase) have demonstrated encouraging efficacy and safety profiles, highlighting the potential of protein degraders as a new class of cancer therapeutics [143,144]. Since NAA10 and its auxiliary subunit NAA15 are overexpressed in certain cancers, protein degraders targeting NAA10 or NAA15 offer a promising strategy to eliminate NAA10 in tumors where its overexpression drives tumorigenesis.

It is important to recognize that targeting NAA10 in cancer presents unique challenges due to its functional duality, tissue-specific roles, and involvement in complex regulatory networks. Although previous studies have shown that NAA10 can function as either an oncogene or tumor suppressor depending on the cancer context, accumulating evidence suggests that it predominantly acts as a cancer promoter in most tumor types. Nevertheless, its precise classification as an oncoprotein or tumor suppressor remains controversial, underscoring the need for a deeper understanding of its context-dependent mechanisms. Second, systemic toxicity is a potential risk due to NAA10′s essential roles in normal development, particularly in cardiac and neuronal functions, as evidenced by the severe phenotypes observed in Ogden syndrome patients with loss-of-function mutations. Addressing the challenge of toxicity will require the development of selective inhibitors or degraders, biomarker-driven strategies, and combinatorial approaches that exploit synthetic lethal interactions.

## 6. Conclusions and Future Perspective

NAA10, a key protein acetyltransferase, is a multifaceted regulatory factor with diverse roles in normal development, human disease, and cancer biology. Through its dual NAT and KAT activities, NAA10 acetylates a substantial portion of the human proteome, influencing numerous cellular processes including proliferation, differentiation, migration, apoptosis, and autophagy. The discovery of *NAA10* mutations in developmental disorders, its alterations in various human cancers, and the correlation between NAA10 overexpression and cancer progression underscore its clinical significance. Studies from the past two decades have provided a deep understanding of NAA10′s structure, regulation, and molecular mechanisms, laying a critical foundation for the development of targeted strategies and inhibitors to modulate its function and activity for potential therapeutic applications in human diseases and cancers.

Despite significant advances in elucidating NAA10′s structure and function, many key questions remain regarding its diverse roles in human biology and disease. Future research should focus on systematically identifying and characterizing NAA10 substrates across different cell types, developmental stages, and disease contexts to clarify how its substrate repertoire and functions change in response to physiological and pathological cues. Dissecting the molecular determinants underlying NAA10′s context-dependent roles—either as an oncogene or tumor suppressor—will be essential, particularly through integrative multi-omics and functional studies in both cancer and developmental models. High-resolution structural studies will aid in designing allosteric or conformationally selective inhibitors with improved specificity and safety. Additionally, the identification of robust biomarkers, such as NAA10 post-translational modification signatures or co-expression with oncogenic drivers, will be critical for patient stratification and precision medicine approaches.

As our understanding of NAA10 biology continues to expand, this multifunctional protein is increasingly recognized as a potential diagnostic marker and therapeutic target in various human diseases, particularly cancer. The pleiotropic functions of NAA10 highlight the complexity of protein acetylation in regulating cellular processes and emphasize the need for continued research in this dynamic and rapidly evolving field.

## Figures and Tables

**Figure 1 cells-14-00863-f001:**
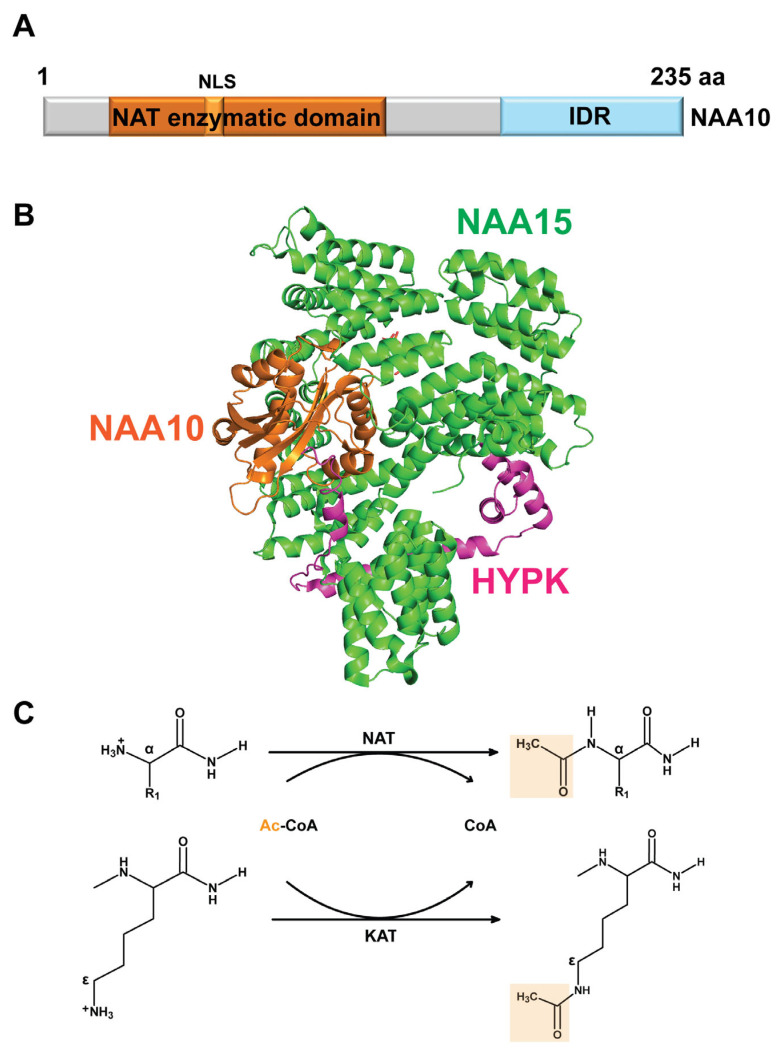
Structure of NAA10 and its catalytic acetylation activity. (**A**) A schematic representation of the NAA10 protein showing the N-terminal NAT enzymatic domain, nuclear localization signal (NLS), and C-terminal intrinsically disordered region (IDR). (**B**) Three-dimensional structure (PDB: 6C95) of the human NatA complex, comprising NAA10 (residues 1–160), NAA15 (residues 5–841), and HYPK (residues 35–119) [41]. The structure was visualized using PyMOL, with NAA10 shown in orange, NAA15 in green, and HYPK in magenta. (**C**) NAA10 functions as a NAT or KAT, catalyzing the transfer of an acetyl group (CH_3_CO) from acetyl-CoA (Ac-CoA) to the free α-amino group at protein N-termini or to the ε-amino group of lysine side chains, respectively. Top: The N-terminal residue is depicted with R1 representing the side chain of small amino acids (Ala, Ser, Thr, Val, Gly, Cys) that are targeted by NAA10 following initiator methionine cleavage. Bottom: A simplified lysine residue is shown to illustrate the KAT activity of NAA10 on internal lysine residues of proteins.

**Figure 2 cells-14-00863-f002:**
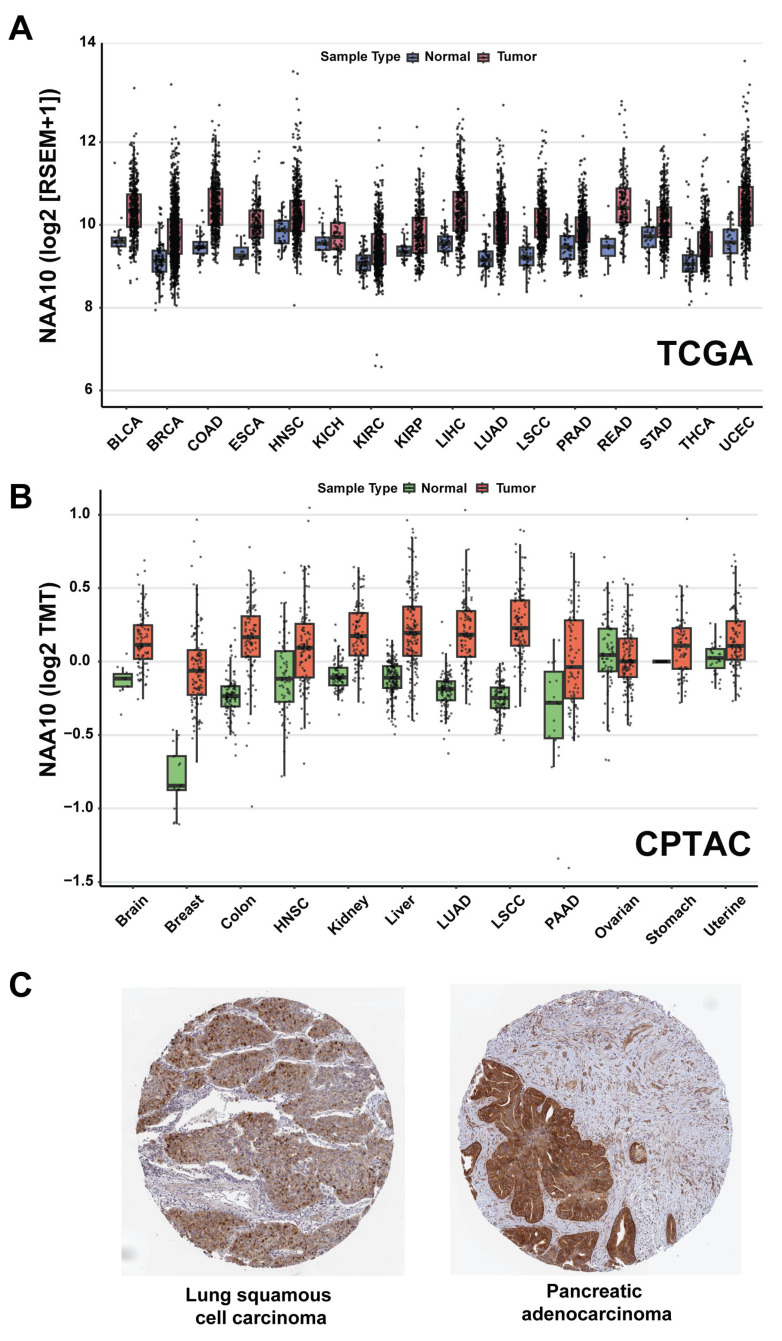
Pan-cancer analysis of NAA10 mRNA and protein expression. (**A**) Boxplots with individual data points show *NAA10* mRNA expression levels (log_2_[RSEM + 1]) in normal and tumor samples across 16 TCGA cancer types. Normalized RNA-sequencing data from TCGA samples were downloaded from the GDC portal (https://gdc.cancer.gov, accessed on 24 April 2025) [96]. (**B**) Boxplots display NAA10 protein expression (log_2_ TMT abundance) in tumor and adjacent normal tissue samples from 12 CPTAC cancer types. Normalized proteomics data from CPTAC samples were downloaded from cProSite (https://cprosite.ccr.cancer.gov/, accessed on 17 December 2024) [95]. For the CPTAC stomach cancer dataset, protein abundance in tumor and matched adjacent normal tissues was measured and analyzed as paired samples in the original study, resulting in all normal controls being normalized to zero in the boxplot. (**C**) Two representative images showing high NAA10 IHC staining scores in lung squamous cell carcinoma (LSCC) and pancreatic adenocarcinoma (PAAD), obtained from the HPA portal (https://www.proteinatlas.org/, accessed on 5 May 2025) [98]. BLCA: bladder urothelial carcinoma; BRCA: breast invasive carcinoma; COAD: colon adenocarcinoma; ESCA: esophageal carcinoma; HNSC: head and neck squamous cell carcinoma: KICH: kidney chromophobe; KIRC: kidney renal clear cell carcinoma; KIRP: kidney renal papillary cell carcinoma; LIHC: liver hepatocellular carcinoma; LUAD: lung adenocarcinoma; PRAD: prostate adenocarcinoma; READ: rectum adenocarcinoma; STAD: stomach adenocarcinoma; THCA: thyroid carcinoma; UCEC: uterine corpus endometrial carcinoma.

**Figure 3 cells-14-00863-f003:**
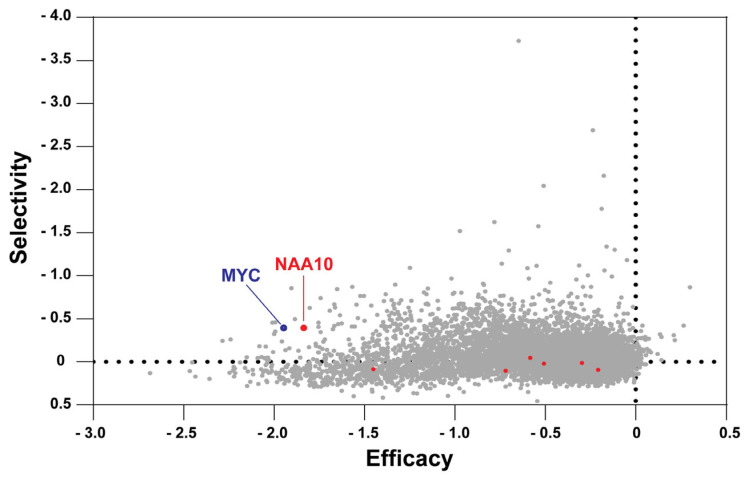
The efficacy and selectivity of NAA genes as shown in the shinyDepMap plot, which integrates CRISPR and shRNA screening data across a panel of tumor cell lines [102]. In a total of 15,847 genes analyzed, each dot (in red, blue, or grey) represents an individual human gene. The colors indicate the following: red represents *NAA10* and six other NAA genes (*NAA20*, *NAA30*, *NAA40*, *NAA50*, *NAA60*, *NAA80*), blue represents the classical oncogene *MYC*, and grey represents all other genes.

**Figure 4 cells-14-00863-f004:**
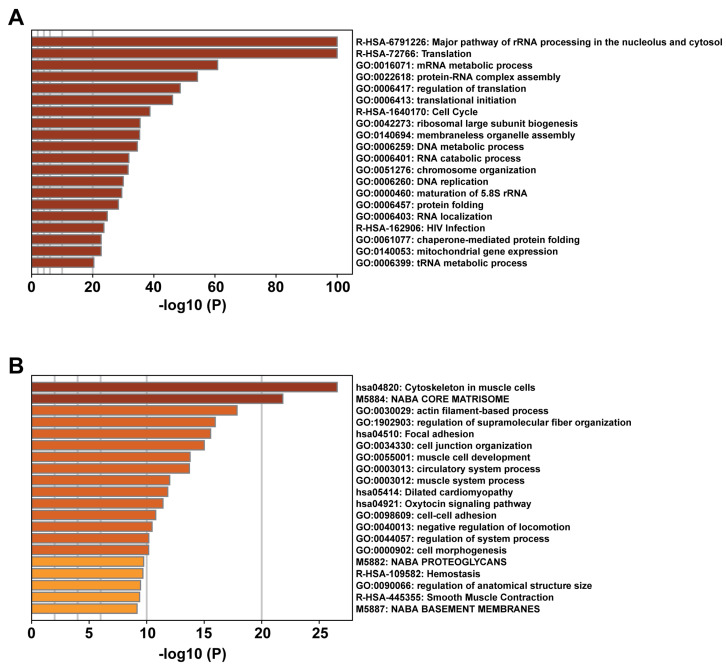
Metascape pathway enrichment analysis of the top proteins that showed strong correlation with NAA10 protein levels in the CPTAC dataset: (**A**) 729 positively correlated proteins and (**B**) 351 negatively correlated proteins [118].

**Figure 5 cells-14-00863-f005:**
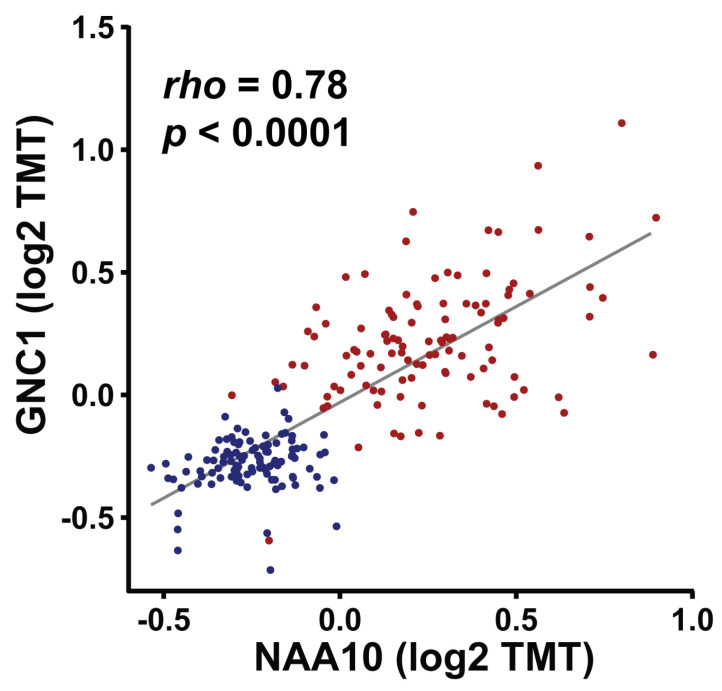
NAA10 is significantly positively correlated with GCN1 in the CPTAC-LSCC dataset [95]. Each dot represents an individual sample, with tumor samples shown in red and adjacent normal tissues in blue. Protein abundances of NAA10 and GCN1 are lower in normal tissues, with data points clustered closely together. In contrast, tumor samples exhibit higher expression of both NAA10 and GCN1, with a greater degree of variability.

**Table 1 cells-14-00863-t001:** Summary of proteins acetylated or interacting with NAA10 identified in published studies.

Modification Type	Target Protein	Lysine Site	PubMed ID	Publication Year
N-terminal acetylation	16 (ELOA, GCN1, PPIA, RPL13A, RPP30, SAE1)		19420222	2009
	TSC2		20145209	2010
	MMP2		29960050	2018
	PLIN2		30425097	2019
	PGC-1α		31422874	2019
	985 (MYH7, MYH6, TNNI3, DES)		40234403	2025
Internal lysine acetylation	HIF-1α	K532	12464182	2002
	β-catenin	N/A	17108104	2006
	MLCK	K608	19826488	2009
	NAA10	K136	20501853	2010
	MSRA	K49	25341044	2014
	RUNX2	K225	25376646	2014
	CDC25A	N/A	26967250	2016
	AR	K618	27659526	2016
	HSP70	K77	27708256	2016
	PGK1	K388	28486006	2017
	AURKA	K75, K125	28915666	2017
	SAMHD1	K405	28978134	2017
	NRF2	K438	36442525	2023
Protein–protein interaction	DNMT1		20592467	2010
	PIX		21295525	2011
	RelA		22496479	2012
	STAT5α		24925029	2014
	ADAM9		32719332	2020
	IKKα		34060226	2021

Note: The numbers of target proteins in studies 19420222 and 40234403 represent the total proteins with altered N-terminal acetylation following NAA10 inhibition. The proteins listed in parentheses are representative examples highlighted in the original studies.

**Table 2 cells-14-00863-t002:** Top candidate NAA10 N-terminally acetylated proteins in cancer from multiple datasets.

Uniport ID	Protein Symbol	Description	NAA10 Meta P
Q9UQ80	PA2G4	proliferation-associated 2G4	30.1
P62913	RPL11	ribosomal protein L11	23.5
P26641	EEF1G	eukaryotic translation elongation factor 1 gamma	21.0
Q7L2H7	EIF3M	eukaryotic translation initiation factor 3 subunit M	21.0
Q14690	PDCD11	programmed cell death 11	21.0
Q12904	AIMP1	aminoacyl tRNA synthetase complex interacting multifunctional protein 1	20.9
Q92616	GCN1	GCN1 activator of EIF2AK4	20.5
Q9UHD1	CHORDC1	cysteine and histidine rich domain containing 1	19.6
P25205	MCM3	minichromosome maintenance complex component 3	17.4
O43583	DENR	density regulated re-initiation and release factor	16.5
Q15007	WTAP	WT1 associated protein	15.9
Q9UL63	MKLN1	muskelin 1	14.9
P62195	PSMC5	proteasome 26S subunit, ATPase 5	14.0
P49790	NUP153	nucleoporin 153	13.5
Q9NPD3	EXOSC4	exosome component 4	13.4
Q9UHB9	SRP68	signal recognition particle 68	13.0
Q9Y5J9	TIMM8B	translocase of inner mitochondrial membrane 8 homolog B	11.6
Q9NR33	POLE4	DNA polymerase epsilon 4, accessory subunit	10.5
Q92990	GLMN	glomulin, FKBP associated protein	10.0
P10412	H1-4	H1.4 linker histone	−10.1
P07305	H1-0	H1.0 linker histone	−10.6

Note: The Meta *p*-value, which integrates correlation results across pan-cancer samples in the CPTAC dataset, was obtained from LinkedOmicsKB. Proteins with Meta-P numbers indicated by “−” show a negative correlation, while the remaining 19 proteins show a positive correlation with NAA10 protein abundance.

## Data Availability

Not applicable.

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
