# Peer review of "NAA10 (N-Alpha-Acetyltransferase 10): A Multifunctional Regulator in Development, Disease, and Cancer"

_cells, 2025, doi:10.3390/cells14120863_

Round 1

Reviewer 1 Report

Comments and Suggestions for Authors

This is a well-organized review and will be a nice fit for the special issue. In table 1, for each entry, the related ref should be cited accordingly. About internal lysine acetylation, there were some controversies in the field and they need to be discussed to enhance the accuracy of the manuscript.

Author Response

First, we appreciate the reviewer’s comment: “This is a well-organized review and will be a nice fit for the special issue.”

In response to the reviewer’s comment, “In Table 1, for each entry, the related reference should be cited accordingly,” we carefully reviewed the references listed in Table 1. During this process, we identified a typographical error in the publication year for PMID: 27708256, which was incorrectly listed as 2014. The correct year is 2016, and this has been corrected in the revised manuscript.

In response to the comment: “Regarding internal lysine acetylation, there are some controversies in the field that should be discussed to enhance the accuracy of the manuscript,” we have added a new paragraph, highlighted in blue, to Section 4.3, which reads as follows:

 “Although NAA10 has been shown to function as a KAT in various studies, particularly in the context of cancer, this activity remains a subject of ongoing debate, as mentioned previously . Structural analyses by Magin et al. (2016) revealed that the NAA10 active site is optimized for N-terminal acetylation, raising questions about its ability to efficiently accommodate internal lysine residues. Additionally, in vitro biochemical assays using recombinant NAA10 failed to acetylate lysine residues within MSRA and RUNX2. Based on these findings, Magin et al. proposed that the observed lysine acetylation of MSRA and RUNX2 may result from non-enzymatic chemical reactions rather than direct enzymatic activity by NAA10. However, a study by Vo et al. (2020) demonstrated that recombinant human NAA10 can acetylate internal lysine residues in vitro, but only under specific, tightly controlled conditions. Notably, the monomeric form of NAA10 exhibited KAT activity, whereas its oligomeric form, which accumulates over time during protein purification, lost this activity [1]. Furthermore, multiple studies using cancer cell models have shown that NAA10 acetylates specific lysine residues on various substrate proteins, as listed in Table 1. Nevertheless, while evidence supports a potential KAT function for NAA10, particularly in cancer models, further biochemical and structural studies are needed to definitively characterize its lysine acetylation activity, as well as its underlying structural and biochemical mechanisms.

We also carefully reviewed all references to ensure they are relevant to the content of the manuscript.

Reviewer 2 Report

Comments and Suggestions for Authors

The review manuscript describes the role of the N-acetyltransferase NAA10 in developmental disorders and cancer. The work is well-written and the content is meaningful and suitable for this journal. The manuscript can become suitable for publication after minor revision:

Figure 1C: The shown amino acid structures suggest that they are not part of a peptide (CONH2, NH-methyl). Please modify the structures. Please specify R1, if possible.

Section 5: How far can other GNAT inhibitors and known HAT inhibitors such as p300 inhibitors become suitable for targeting and inhibiting NAA10?

Section 5: The authors described the PPI example of MDM2-p53 which can be targeted by drugs. Maybe the authors can also provide information about crucial PPIs of acetyl transferases (e.g., p300) and inhibition of HAT-involving PPIs (e.g., inhibition of p300-MYB PPI) with small-molecule inhibitors. Are drugs available which inhibit NAA10 protein partners (DNMT1, IKKα, etc.)?

Author Response

Comments #1 “The review manuscript describes the role of the N-acetyltransferase NAA10 in developmental disorders and cancer. The work is well-written and the content is meaningful and suitable for this journal.”

Response #1: First, we thank the reviewer for this positive comment. We also appreciate the additional insightful feedback and constructive suggestions, which have helped to significantly improve the quality of our revised manuscript.

Comments #2 "Figure 1C: The shown amino acid structures suggest that they are not part of a peptide (CONH2, NH-methyl). Please modify the structures. Please specify R1, if possible."

Response #2 : We respectfully agree with the reviewer’s comment. To address this concern, we have updated the figure legend to clarify the structural context, although we have not modified the chemical structures in Figure 1C. The following sentences have been added to the legend:

“Top: The N-terminal residue is depicted with R1 representing the side chain of small amino acids (Ala, Ser, Thr, Val, Gly, Cys) that are targeted by NAA10 following initiator methionine cleavage. Bottom: A simplified lysine residue is shown to illustrate the KAT activity of NAA10 on internal lysine residues of proteins.”

We chose not to alter the structures in the Figure 1C because the simplified representations facilitate visualization of the enzymatic concepts.

Comments #3 "Section 5: How far can other GNAT inhibitors and known HAT inhibitors such as p300 inhibitors become suitable for targeting and inhibiting NAA10?"

Response #3: We thank the reviewer for raising this important question regarding the potential to repurpose existing GNAT and KAT inhibitors, such as EP300 inhibitors, for targeting NAA10. In response, we have revised Section 5 to include a more detailed discussion of the opportunities and challenges associated with this approach. Specifically, we address the structural and functional distinctions between NAA10 and other GNAT/KAT family members, as well as differences in substrate specificity, which may influence inhibitor selectivity. We have added the following text to the revised manuscript.

As described previously, NAA10 belongs to the GNAT family within the larger KAT superfamily, which includes more than 30 members in humans, such as KAT2A, KAT2B, KAT6A, KAT6B, EP300, and CREBBP (CREB binding lysine acetyltransferase) [1-3,31,121,122]. These KATs are frequently altered in various human cancers, through mutations, translocations, gene amplification, or overexpression; therefore, inhibitors targeting them have been extensively investigated [31,121,123-125]. Over the past two decades, various approaches have been used to develop inhibitors targeting KAT2A (GCN5), but their potency remains suboptimal [125-127]. In 2017, Lasko et al. overcame the long-standing challenge of developing a drug-like KAT inhibitor by identifying A-485, a first-in-class, highly potent, selective, cell and in vivo active EP300/CREBBP catalytic inhibitor [126]. A-485 has been shown to inhibit the proliferation of various cancer cell lines, including those from prostate cancer, melanoma, multiple myeloma, and non-Hodgkin lymphoma [123,128-130]. More recently, catalytic inhibitors of KAT6A and KAT6B have been identified [125,131-133]. One such compound, PF-07248144, demonstrated a tolerable safety profile and durable antitumor activity in heavily pretreated estrogen receptor-positive (ER+), metastatic breast cancer in a Phase 1 trial (NCT04606446) [133]. These successfully identified KAT inhibitors, particularly those targeting KAT6, have established KATs as druggable targets in cancer. Although NAA10 shares the conserved GNAT fold with other GNAT enzymes, suggesting potential overlap in inhibitor binding, its unique role in the NatA complex, along with its dual NAT and debated KAT activities, reflects distinct substrate specificity and active site architecture compared to other GNATs or KATs such as KAT6 or EP300/CREBBP [41,45,121-123,125,127]. Nevertheless, further studies are needed to assess the cross-reactivity of currently available KAT catalytic inhibitors, and to explore the potential of structure-guided drug design for developing selective NAA10 inhibitors."

Comments #4 "Section 5: The authors described the PPI example of MDM2-p53 which can be targeted by drugs. Maybe the authors can also provide information about crucial PPIs of acetyl transferases (e.g., p300) and inhibition of HAT-involving PPIs (e.g., inhibition of p300-MYB PPI) with small-molecule inhibitors. Are drugs available which inhibit NAA10 protein partners (DNMT1, IKKα, etc.)?"

Response #4: We thank the reviewer for this insightful comment. Indeed, small-molecule inhibitors targeting KAT-involved protein–protein interactions (PPIs) have been reported. In addition, drugs targeting NAA10-interacting partners, such as DNMT1, are already available and have been used clinically. We have integrated this information into Section 5 of the revised manuscript. We have added the following text to the revised manuscript.

Previous studies have also demonstrated therapeutic potential of inhibitors targeting KAT-involved PPIs. For example, small molecules that disrupt the EP300–MYB interaction inhibit acute myeloid leukemia (AML) cell proliferation both in vitro and in vivo [136,137]. Additionally, multiple EP300/CREBBP inhibitors targeting the bromodomain-mediated PPIs of EP300/CREBBP have been identified and clinically developed. Several of these compounds, including Inobrodib (CCS1477), FT-7051, and NEO2734 , have entered clinical trials for the treatment of various cancers, such as leukemia, lymphoma, breast, lung, and prostate cancers [123,125,138-140]. Biochemical and structural studies have demonstrated that the interaction between NAA10 and NAA15 is critical for NatA complex activity [17,62]. Given detailed understanding of the interaction interface and key residues involved, it is plausible to hypothesize that inhibitors targeting the NAA10 - NAA15 interaction could be identified and developed as a novel therapeutic strategy for cancer treatment. Furthermore, NAA10 also interacts with other proteins, such as DNMT1. FDA-approved DNMT1 inhibitors, including decitabine and azacitidine, are currently used to treat myelodysplastic syndromes; however, these drugs target the enzymatic activity of DNMT1 rather than its interaction with NAA10 [141,142]. In the future, it would be worthwhile to investigate whether these DNMT1 inhibitors have therapeutic potential in tumors with NAA10 overexpression."

All changes made in response to the reviewers’ feedback are highlighted in blue in the revised manuscript.

Finally, thank you again for your insightful comments and constructive suggestions.